# On the Onset of Robust Overfitting in Adversarial Training

## Abstract

Adversarial Training (AT) is a widely-used algorithm for building robust neural networks, but it suffers from the issue of robust overfitting, the fundamental mechanism of which remains unclear. In this work, we consider normal data and adversarial perturbation as separate factors, and identify that the underlying causes of robust overfitting stem from the normal data through factor ablation in AT. Furthermore, we explain the onset of robust overfitting as a result of the model learning robust features that lack generalization, which we refer to as non-effective features. Specifically, we offer a detailed analysis of how the robustness gap between the training and test sets prompts the generation of non-effective features, ultimately leading to robust overfitting. Additionally, we analysis various empirical behaviors observed in robust overfitting and revisit different techniques to mitigate robust overfitting from the perspective of non-effective features, providing a comprehensive understanding of the robust overfitting. This understanding inspires us to propose two measures, attack strength and data augmentation, to hinder the learning of non-effective features by the neural network, thereby alleviating robust overfitting. Besides, extensive experiments conducted on benchmark datasets demonstrate the effectiveness of the proposed methods in enhancing adversarial robustness.

## 1 Introduction

Adversarial Training (AT) (Madry et al., 2018) has emerged as a reliable method for improving a model's robustness against adversarial attacks (Szegedy et al., 2014; Goodfellow et al., 2015). It involves training networks using adversarial data generated on-the-fly and has been proven to be one of the most effective empirical defenses (Athalye et al., 2018). AT has shown success in building robust neural networks when applied to the MNIST dataset. However, achieving the same goal on more complex datasets like CIFAR10 has proven to be challenging (Madry et al., 2018). Apart from the limited capacity of current neural networks (Nakkiran, 2019), there is also a perplexing phenomenon known as robust overfitting (Rice et al., 2020) that significantly hampers this process. Specifically, when robust overfitting occurs during AT, the model's robust accuracy on test data continues to decline with further training. This phenomenon has been observed across different datasets, network architectures, and AT variants (Rice et al., 2020).

Recently, various technologies have been proposed to empirically alleviate robust overfitting (Carmon et al., 2019; Chen et al., 2020b; Dong et al., 2022; Wu et al., 2020; Yu et al., 2022b). For instance, Wu et al. (2020) proposed the double-perturbation mechanism, which adversarially perturbs both inputs and weights to achieve a smoother weight-loss landscape. Yu et al. (2022b) introduced the Minimum Loss Constrained Adversarial Training (MLCAT) prototype to prevent the model from fitting the small-loss adversarial data. Both methods can alleviate robust overfitting while enhancing adversarial robustness. However, the essential issue, the fundamental mechanism behind robust overfitting, remains unresolved and is of critical importance.

In this paper, we investigate the fundamental mechanism of robust overfitting. Firstly, we show that the inducing factors of robust overfitting stem from normal data. Specifically, we treat normal data and adversarial perturbations as separate factors, and devise factor ablation adversarial training to assess their respective impacts on robust overfitting. We observe that simultaneously ablating adversarial perturbations and normal data in adversarial training can greatly mitigate the robust

overfitting, whereas adversarial training that only ablates adversarial perturbations still exhibits a severe degree of robust overfitting. Given that these experiments strictly adhere to the principle of controlling variables, with the sole difference being the presence of normal data in the training set, we can infer that the underlying causes of robust overfitting stem from normal data.

Normal data can be regarded as a composition of features. To gain more insights into the mechanism of robust overfitting, we provide a detail analysis for the onset of robust overfitting in adversarial training from the perspective of feature generalization. To begin with, due to the distribution deviation of normal data between training and test sets, certain non-generalizable robust features emerge in the training set, which we denote as non-effective features. Subsequently, during the adversarial training process, the model's adversarial robustness on the training set exceeds that on the test set. This results in a robustness gap between the training and test data. Considering that adversarial perturbations are generated on-the-fly and adaptively adjusted based on the model's robustness, the robustness gap between the training and test data leads to distinct adversarial perturbation on the robust features in these datasets. The varying degree of adversarial perturbation on the robust features in the training and test data amplify the distribution differences between these two datasets, thereby degrading the generalization of robust features on the training set and facilitating the generation of non-effective features. The increasing non-effective features further exacerbates the robustness gap between training and test data, forming a vicious cycle. As adversarial training advances, the robustness gap between the training and test sets progressively expands, promoting the generation of non-effective features. When the model's optimization is govern by these non-effective features, it results in the phenomenon of robust overfitting. Correspondingly, we provide a comprehensive explanation for various empirical behaviors associated with robust overfitting and revisit different existing techniques for mitigating robust overfitting based on our analysis.

In order to support our analysis, we also devise two representative measures, namely attack strength and data augmentation, to regulate the model's learning of non-effective features. Specifically, *i)* eliminating non-effective features through adversarial perturbations; and *ii)* aligning the model's adversarial robustness on the training set with that on the test set through data augmentation techniques. Both measures provide a flexible way to control the generation of non-effective features. We observe a clear correlation between the extent of robust overfitting and the model's learning of non-effective features: the fewer non-effective features the model learns, the less pronounced the degree of robust overfitting. These findings align well with our analysis. Furthermore, extensive experiments conducted in a wide range of settings also validate the effectiveness of the proposed measures in enhancing adversarial robustness. To sum up, our contributions are as follows:

- We conducted a series of rigorously factor ablation experiments following the principles of the controlled variable method, inferring that the factors inducing robust overfitting originate from normal data.
- We provide a comprehensive understanding of the onset robust overfitting through a detailed analysis of the generation of non-effective features.
- Based on the understanding, we devise two representative measures to impede the model's learning of non-effective features, validating our analysis. Moreover, extensive experiments demonstrate that the proposed methods consistently enhance the adversarial robustness of baseline methods by a noticeable margin.

## 2 RELATED WORK

In this section, we briefly review related literature from two perspectives: adversarial training and robust overfitting.

### 2.1 ADVERSARIAL TRAINING

Let $f_\theta$, $\mathcal{X}$ and $\ell$ represent the neural network $f$ with model parameter $\theta$, the input space, and the loss function, respectively. Given a $C$-class dataset $\mathcal{S} = \{(x_i, y_i)\}_{i=1}^n$, where $x_i \in \mathcal{X}$ and $y_i \in \mathcal{Y} = \{0, 1, \ldots, C-1\}$ denotes its corresponding label, the objective function of *standard training* is

$$\min_\theta \frac{1}{n} \sum_{i=1}^n \ell(f_\theta(x_i), y_i), \tag{1}$$

where the neural network $f_\theta$ learns features in $x_i$ that are correlated with associated labels $y_i$ in order to minimize the empirical risk of misclassifying normal inputs. However, empirical evidence (Szegedy et al., 2014; Tsipras et al., 2018; Ilyas et al., 2019) suggests that networks trained under this regime tend to fit fragile, non-robust features that are incomprehensible to humans. To address this issue, adversarial training introduces adversarial perturbations to each data point by transforming $\mathcal{S} = \{(x_i, y_i)\}_{i=1}^n$ into $\mathcal{S}' = \{(x_i' = x_i + \delta_i, y_i)\}_{i=1}^n$. The adversarial perturbations $\{\delta_i\}_{i=1}^n$ are constrained by a pre-specified budget, *i.e.* $\{\delta \in \Delta : ||\delta||_p \le \epsilon\}$, where $p$ can be $1, 2, \infty$, etc. Therefore, the objective function for *adversarial training* (Madry et al., 2018) is

$$\min_\theta \frac{1}{n} \sum_{i=1}^n \max_{\delta_i \in \Delta} \ell(f_\theta(x_i + \delta_i), y_i), \tag{2}$$

where the inner maximization process generates adversarial perturbations on-the-fly that maximizes the classification loss. Subsequently, the outer minimization process optimizes the neural network using the generated adversarial data. This iterative procedure aims to achieve an adversarially robust classifier. The most commonly employed approach for generating adversarial perturbations in AT is Projected Gradient Descent (PGD) (Madry et al., 2018), which applies adversarial attack to normal data $x_i$ over multiple steps $k$ with a step size of $\alpha$:

$$\delta^k = \Pi_\Delta(\alpha \cdot \text{sign}\nabla_x \ell(f_\theta(x + \delta^{k-1}), y) + \delta^{k-1}), k \in \mathbb{N}, \tag{3}$$

where $\delta^k$ represents the adversarial perturbation at step $k$, and $\Pi_\Delta$ denotes the projection operator.

Besides the standard AT, there exist several other common variants of adversarial training methods (Kannan et al., 2018; Zhang et al., 2019; Wang et al., 2019). One typical example is TRADES (Zhang et al., 2019), which proposes a regularized surrogate loss that balances natural accuracy and adversarial robustness:

$$\min_\theta \sum_i \left\{ \text{CE}(f_\theta(x_i), y_i) + \beta \cdot \max_{\delta_i \in \Delta} \text{KL}(f_\theta(x_i)||f_\theta(x_i + \delta_i)) \right\}, \tag{4}$$

where CE is the cross-entropy loss that encourages the network to maximize natural accuracy, KL is the Kullback-Leibler divergence that encourages improvement of robust accuracy, and the hyperparameter $\beta$ is employed to regulates the tradeoff between natural accuracy and adversarial robustness.

## 2.2 Robust Overfitting

Robust overfitting was initially observed in standard AT (Madry et al., 2018). Later, Rice et al. (2020) conducted a comprehensive study and discovered that conventional remedies used for overfitting in deep learning are of little help in combating robust overfitting in AT. This finding prompted further research efforts aimed at mitigating robust overfitting. Schmidt et al. (2018) attributed robust overfitting to sample complexity theory and suggested that more training data are required for adversarial robust generalization, which is supported by empirical results in derivative works (Carmon et al., 2019; Alayrac et al., 2019; Zhai et al., 2019). Recent works also proposed various strategies to mitigate robust overfitting without relying on additional training data, such as sample reweighting (Wang et al., 2019; Zhang et al., 2020; Liu et al., 2021), label smoothing (Izmailov et al., 2018), stochastic weight averaging (Chen et al., 2020b), temporal ensembling (Dong et al., 2022), knowledge distillation (Chen et al., 2020b), weight regularization (Liu et al., 2020; Wu et al., 2020; Yu et al., 2022a;b), and data augmentation (Tack et al., 2022; Li & Spratling, 2023). While these techniques can assist in mitigating robust overfitting, the fundamental mechanism behind robust overfitting remains unclear. This uncertainty has somewhat constrained the widespread applicability of current techniques. For instance, it has been noted that more training data does not necessarily alleviate robust overfitting and can even harm robust generalization (Chen et al., 2020a; Min et al., 2021). Additionally, it was shown that sample reweighting techniques with completely opposing objectives can both effectively alleviate robust overfitting (Zhang et al., 2020; Yu et al., 2022b), and data augmentation technique was found to be inadequate in combating robust overfitting in prior attempts (Gowal et al., 2020; Rebuffi et al., 2021). These contradictions are common in the adversarial training community and further emphasize the importance of understanding the mechanism of robust overfitting. In this work, we investigate the onset of robust overfitting and explore its underlying mechanism.

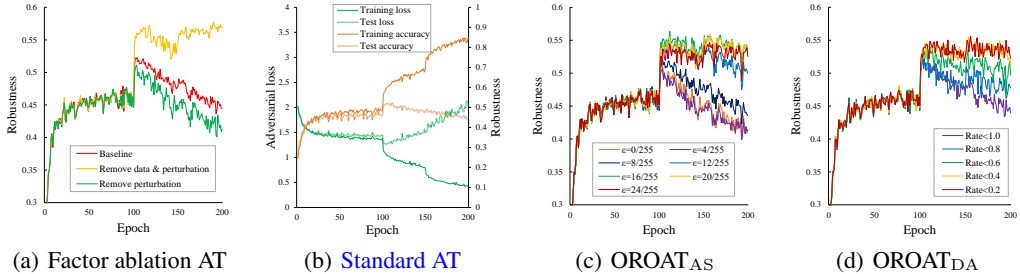

Figure 1: (a) The test robustness of different experimental groups in factor ablation adversarial training; (b) the adversarial loss and robustness of standard AT; (c) the test robustness of OROAT$_{AS}$ with varying attack strengths, and (d) the test robustness of OROAT$_{DA}$ with different proportions of small-loss adversarial data.

## 3 THE ONSET OF ROBUST OVERFITTING IN ADVERSARIAL TRAINING

In this section, we commence with factor ablation experiments to identify the underlying causes of robust overfitting (Section 3.1). Subsequently, we offer an intuitive analysis for the onset of robust overfitting through the lens of feature generalization. To this end, we explain various empirical behaviors associated with robust overfitting and revisit existing techniques for mitigating robust overfitting based on our analysis (Section 3.2). Finally, we develop two representative measures to support our analysis (Section 3.3).

### 3.1 FACTOR ABLATION ADVERSARIAL TRAINING

Inspired by the data ablation experiments in Yu et al. (2022b), which revealed that small-loss adversarial data leads to robust overfitting by removing adversarial data during training, we propose factor ablation adversarial training to gain deeper insights into robust overfitting. We follow the same rule as the data ablation experiments, using a fixed loss threshold to differentiate between large-loss and small-loss adversarial data. For instance, in the CIFAR10 dataset, data with an adversarial loss of less than 1.5 are regarded as small-loss adversarial data. Unlike the data ablation experiments , where adversarial data is treated as a unified entity, we treated normal data and adversarial perturbations within small-loss adversarial data as separate factors and conducted more detailed factor ablation experiments to identify the inducing factor of robust overfitting. Specifically, we trained a PreAct ResNet-18 model on CIFAR-10 using standard AT under the $\ell_\infty$ threat model and removed specified factors before robust overfitting occurred (i.e., at the 100th epoch), including: *i)* **baseline**, which is a baseline group without removing any factors; *ii)* **data & perturbation**, which removes both the normal data and adversarial perturbations from small-loss adversarial data; and *iii)* **perturbation**, which only removes the adversarial perturbations from small-loss adversarial data.

It's important to note that the experimental groups mentioned above were entirely identical before the occurrence of robust overfitting. This ensures that these experiments adhered to a rigorous controlled variable principle, with the only difference between the various experimental groups being the specific factors removed from the training data at the 100th epoch. The experimental results of factor ablation adversarial training are summarized in Figure 1(a). We observe that the **data & perturbation** group exhibits a significant relief in robust overfitting, while both the **baseline** and **perturbation** groups still experience severe robust overfitting. Since the only difference between the **data & perturbation** group and the **perturbation** group is the presence of normal data in the training set, we can clearly infer that normal data is the inducing factor of robust overfitting. Similar effects were also observed across different datasets, network architectures, and adversarial training variants (see Appendix A), indicating that this is a general finding in adversarial training.

### 3.2 THE ONSET OF ROBUST OVERFITTING

In this part, we delve into the analysis of how normal data contributes to robust overfitting from the perspective of feature generalization. Building on this understanding, we further explain various

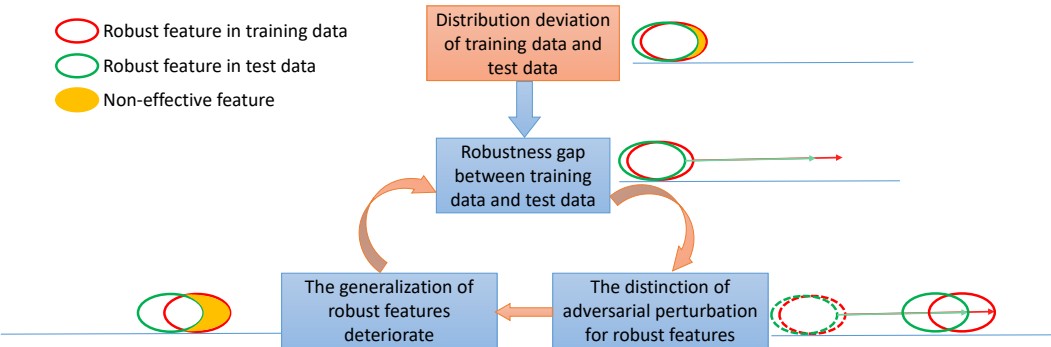

Figure 2: Illustration of the onset of robust overfitting.

empirical behaviors associated with robust overfitting and revisit existing techniques for mitigating robust overfitting.

From the experimental results in Section 3.1, we can know that the factors inducing robust overfitting originate from normal data. Normal data can be considered as a composition of features. According to Ilyas et al. (2019), these features can be further categorized into robust features and non-robust features. Specifically, given a model and a specified attack budget, if the correlation between a feature and its corresponding label consistently holds under the specified attack budget, then this feature is considered robust; otherwise, it is classified as non-robust. However, due to the distribution deviation between the training and test datasets, some robust features in the training set may lack generalization. We refer to these features as non-effective features. Next, we proceed to further analyze the generation of non-effective features during the adversarial training process.

During adversarial training, adversarial perturbations are generated on-the-fly and adaptively adjusted based on the model's robustness. In the initial stages of adversarial training, due to the similarity in the model's robustness between the training and test datasets, the generated adversarial perturbations on the robust features are relatively close in these datasets. Thus, most robust features in the training set can still exhibit good generalization, thereby enhancing the model's adversarial robustness on the test set. However, as training progresses, the model's robustness in the training dataset increases significantly faster than its robustness on the test dataset, as evidenced in Figure 1(b) by the adversarial loss and robustness observed on both datasets. This leads to a widening robustness gap between the training and test datasets. Consequently, the generated adversarial perturbation on the robust features becomes progressively more distinct in these datasets, which degrades the generalization of robust features on the training set and facilitates the generation of non-effective features, causing the model to learn an increasing number of non-effective features. As non-effective features proliferate, the robustness gap between the training and test sets continues to grow, forming a vicious cycle, as illustrated in Figure 2. Once the model's optimization is governed by these non-effective features, the model's adversarial robustness on the test dataset will continue to decline. This, in turn, gives rise to the phenomenon of robust overfitting.

**Empirical behaviors of robust overfitting.** We notice that robust overfitting exhibits some empirical behaviors in adversarial training: 1) Removing small-loss adversarial data can prevent robust overfitting. 2) As the adversarial perturbation budget increases, the degree of robust overfitting initially rises and then decreases. Our analysis naturally explains these phenomena: 1) adversarial data with small loss indicates that the model's robustness on these data is good, maintaining a substantial gap compared to the model's robustness on the test set. The robustness gap promotes the generation of non-effective features on these data. Therefore, removing small-loss adversarial data from the training set can decrease the generation of non-effective features, effectively mitigating robust overfitting. 2) As the perturbation budget increases from 0, the distinction of adversarial perturbation for robust features in the training and test datasets gradually expands, resulting in a higher likelihood of non-effective features generation during training. This explains why natural training does not exhibit robust overfitting, and as the adversarial perturbation budget increases, the degree of robust overfitting also rises. However, with a further increase in the perturbation budget, the degree of robust overfitting decreases. This is because the model's robustness on the training set itself is limited under a large perturbation budget, narrowing the robustness gap between the training and test sets. This reduction in the robustness gap alleviates the generation of non-effective features in the training

---

**Algorithm 1** $\text{OROAT}_{\text{AS}}$

---

1: **Input:** Network $f_\theta$, training data $\mathcal{S}$, mini-batch $\mathcal{B}$, batch size $n$, learning rate $\eta$, PGD step size $\alpha$, PGD budget $\epsilon$, PGD steps $K$, adjusted PGD budget $\epsilon_a$, adjusted PGD steps $K_a$, small-loss data threshold $t$.
2: **Output:** Adversarially robust model $f_\theta$.
3: **repeat**
4:     Read mini-batch $(x_\mathcal{B}, y_\mathcal{B})$ from training set $\mathcal{S}$.
5:     $x'_\mathcal{B} \leftarrow x_\mathcal{B} + \delta$, where $\delta \sim \text{Uniform}(-\epsilon, \epsilon)$
6:     **for** $k = 1$ **to** $K$ **do**
7:         $x'_\mathcal{B} \leftarrow \Pi_\epsilon(x'_\mathcal{B} + \alpha \cdot \text{sign}(\nabla_{x'_\mathcal{B}} \ell(f_\theta(x'_\mathcal{B}), y_\mathcal{B})))$
8:     **end for**
9:     $x = x_\mathcal{B}(\ell(f_\theta(x'_\mathcal{B}), y_\mathcal{B}) \leq t)$
10:    $y = y_\mathcal{B}(\ell(f_\theta(x'_\mathcal{B}), y_\mathcal{B}) \leq t)$
11:    $x' \leftarrow x + \delta$, where $\delta \sim \text{Uniform}(-\epsilon_a, \epsilon_a)$
12:    **for** $k = 1$ **to** $K_a$ **do**
13:        $x' \leftarrow \Pi_{\epsilon_a}(x' + \alpha \cdot \text{sign}(\nabla_{x'} \ell(f_\theta(x'), y)))$
14:    **end for**
15:    $x'_\mathcal{B}(\ell(f_\theta(x'_\mathcal{B}), y_\mathcal{B}) \leq t) = x'$
16:    $\theta \leftarrow \theta - \eta \nabla_\theta \frac{1}{n} \sum_{i=1}^{n} \ell(f_\theta(x'^{(i)}_\mathcal{B}), y^{(i)}_\mathcal{B})$
17: **until** training converged

---

data. Consequently, the degree of robust overfitting gradually decreases. Due to space constraints, we provide more analysis of existing techniques for mitigating robust overfitting in Appendix B.

### 3.3 THE PROPOSED METHODS

As mentioned in Section 3.2, we analyse the Onset of Robust Overfitting in Adversarial Training (OROAT) as a result of learning non-effective features, which are derived from the model's robustness gap between the training and test sets. In this part, we introduce two approaches to support our analysis: *attack strength*, which belongs to the feature-elimination approach, and *data augmentation*, which belongs to the robustness-alignment approach. These two methods are representative and, more importantly, orthogonal in regulating the learning of non-effective features during training, thereby fully validating our analysis of OROAT.

**OROAT through attack strength.** The feature-elimination approach reduces the model's learning of non-effective features by eliminating them from the training dataset. Adversarial training (Goodfellow et al., 2015; Madry et al., 2018) is the primitive method in this direction. It utilizes adversarial perturbations to eliminate the non-robust features in the training dataset.

To achieve different degrees of non-effective feature elimination, we applied varying levels of attack strength to generate adversarial perturbations, thereby adjusting the generalization of the robust features learned by the model. Specifically, we trained PreAct ResNet-18 on CIFAR10 under the $\ell_\infty$ threat model and used different perturbation budgets $\epsilon$ on small-loss adversarial data, ranging from $0/255$ to $24/255$. In each setting, we evaluated the robustness on CIFAR10 test data under the standard perturbation budget of $\epsilon = 8/255$. The pseudocode is provided in Algorithm 1. This approach utilizes the attack strength strategy to eliminate different levels of non-effective features, referred to as $\text{OROAT}_{\text{AS}}$.

The results of $\text{OROAT}_{\text{AS}}$ with different attack strengths are summarized in Figure 1(c). We observe a clear correlation between the applied attack strength and the extent of robust overfitting. Specifically, the more non-effective features are eliminated, the milder the degree of robust overfitting. When the perturbation budget is $0/255$, robust overfitting is most pronounced. However, when the perturbation budget exceeds a certain threshold, such as $16/255$, the model exhibits almost no robust overfitting. It is worth noting that similar patterns are also observed across different datasets, network architectures, and adversarial training variants (as shown in Appendix C). These experimental results clearly demonstrate that robust overfitting is driven by these non-effective features.

**OROAT through data augmentation.** On the other hand, the robustness-alignment approach prevents the generation of non-effective features by aligning the model's robustness between the training and test datasets. The double-perturbation mechanism (Wu et al., 2020) and minimum loss

---

**Algorithm 2** OROAT$_{\text{DA}}$

---

1: **Input:** Network $f_\theta$, training data $\mathcal{S}$, mini-batch $\mathcal{B}$, batch size $n$, learning rate $\eta$, PGD step size $\alpha$, PGD budget $\epsilon$, PGD steps $K$, small-loss data proportion $p$, small-loss data threshold $t$.
2: **Output:** Adversarially robust model $f_\theta$.
3: **repeat**
4:     Read mini-batch $(x_\mathcal{B}, y_\mathcal{B})$ from training set $\mathcal{S}$.
5:     $x'_\mathcal{B} \leftarrow x_\mathcal{B} + \delta$, where $\delta \sim \text{Uniform}(-\epsilon, \epsilon)$
6:     **for** $k = 1$ **to** $K$ **do**
7:         $x'_\mathcal{B} \leftarrow \Pi_\epsilon(x'_\mathcal{B} + \alpha \cdot \text{sign}(\nabla_{x'_\mathcal{B}} \ell(f_\theta(x'_\mathcal{B}), y_\mathcal{B})))$
8:     **end for**
9:     $n_t = \sum \mathbb{I}(\ell(f_\theta(x'_\mathcal{B}), y_\mathcal{B}) \leq t)$
10:     **if** $n_t/n > p$ **then**
11:         **repeat**
12:             $x = x_\mathcal{B}(\ell(f_\theta(x'_\mathcal{B}), y_\mathcal{B}) \leq t)$
13:             $y = y_\mathcal{B}(\ell(f_\theta(x'_\mathcal{B}), y_\mathcal{B}) \leq t)$
14:             $x = \text{DataAugmentation}(x)$
15:             $x' \leftarrow x + \delta$, where $\delta \sim \text{Uniform}(-\epsilon, \epsilon)$
16:             **for** $k = 1$ **to** $K$ **do**
17:                 $x' \leftarrow \Pi_\epsilon(x' + \alpha \cdot \text{sign}(\nabla_{x'} \ell(f_\theta(x'), y)))$
18:             **end for**
19:             $x'_\mathcal{B}(\ell(f_\theta(x'_\mathcal{B}), y_\mathcal{B}) \leq t)(\ell(f_\theta(x'), y) > t) = x'(\ell(f_\theta(x'), y) > t)$
20:             $n_t = \sum \mathbb{I}(\ell(f_\theta(x'_\mathcal{B}), y_\mathcal{B}) \leq t)$
21:         **until** $n_t/n \leq p$
22:     **end if**
23:     $\theta \leftarrow \theta - \eta \nabla_\theta \frac{1}{n} \sum_{i=1}^n \ell(f_\theta(x'^{(i)}_\mathcal{B}), y^{(i)}_\mathcal{B})$
24: **until** training converged

---

constraint (Yu et al., 2022b) are the foundational methods in this direction, which utilize weight perturbations to diminish the model's robustness specifically on the training set.

During AT, the model's robustness on the training set typically exceeds that on the test set. For a given sample, we can roughly estimate the model's robustness by examining the adversarial loss associated with that sample. For instance, a larger adversarial loss indicates that the model possesses weaker robustness on the given sample. To conduct AT with varying model robustness on the training set, we employ data augmentation techniques to adjust the proportion of small-loss adversarial data in each minibatch. Specifically, at the beginning of each iteration, we check whether the proportion of small-loss adversarial data meets the specified threshold. If this proportion is below the specified threshold, we apply data augmentation to these small-loss examples within the minibatch until the desired proportion is reached. The pseudocode is provided in Algorithm 2, where the data augmentation method we use is AugMix (Hendrycks et al., 2020). We refer to this adversarial training framework, empowered by the data augmentation technique, as OROAT$_{\text{DA}}$.

The results of OROAT$_{\text{DA}}$ with different proportions of small-loss adversarial data are summarized in Figure 1(d). We observe a clear correlation between the proportion of small-loss data and the extent of robust overfitting. As the model's robustness on the training dataset decreases, the degree of robust overfitting becomes increasingly mild. Furthermore, these effects are consistent across different datasets, network architectures, and adversarial training variants (as shown in Appendix D). These empirical results strongly support our analysis of OROAT, indicating that the model's robustness gap promotes the generation of non-effective features.

## 4 EXPERIMENT

In this section, we evaluate the effectiveness of the proposed methods. Section 4.1 demonstrates that both OROAT$_{\text{AS}}$ and OROAT$_{\text{DA}}$ consistently enhance the adversarial robustness compared to the baselines. In Section 4.2, we conduct the analysis and discussion regarding the proposed methods.

**Setup.** We conduct extensive experiments across different benchmark datasets (CIFAR10 and CIFAR100 (Krizhevsky et al., 2009)), network architectures (PreAct ResNet-18 (He et al., 2016) and Wide ResNet-34-10 (Zagoruyko & Komodakis, 2016)), and adversarial training variants (AT (Madry et al., 2018) and TRADES (Zhang et al., 2019)). In addition to adversarial training variants, we also

Table 1: Evaluation of adversarial robustness for $OROAT_{AS}$ and $OROAT_{DA}$. The results were calculated as the average of three random trials. We omit the standard deviations as they are small (Natural$< 0.6\%$, PGD-20$< 0.3\%$ and AA$< 0.2\%$).

| Network | Dataset | Method | Best | | | Last | | |
|---|---|---|---|---|---|---|---|---|
| | | | Natural | PGD-20 | AA | Natural | PGD-20 | AA |
| PreAct ResNet-18 | CIFAR10 | AT | 82.31 | 52.28 | 48.09 | 84.11 | 44.46 | 42.01 |
| | | $OROAT_{DA}$ | **82.58** | 53.95 | 48.48 | **85.45** | 49.69 | 44.44 |
| | | $OROAT_{AS}$ | 77.68 | **56.37** | **49.37** | 78.04 | **51.96** | **45.97** |
| | CIFAR100 | AT | 55.14 | 28.93 | 24.53 | 55.83 | 20.87 | 18.92 |
| | | $OROAT_{DA}$ | **55.79** | 29.40 | 24.80 | **57.92** | 25.51 | 21.59 |
| | | $OROAT_{AS}$ | 51.02 | **30.25** | **25.63** | 51.06 | **26.19** | **22.67** |
| | CIFAR10 | TRADES | 81.50 | 52.92 | 48.90 | 82.27 | 49.95 | 46.92 |
| | | $OROTRADES_{DA}$ | **82.89** | 53.14 | 49.12 | **83.28** | **52.13** | 48.41 |
| | | $OROTRADES_{AS}$ | 80.92 | **53.49** | **49.88** | 80.97 | 52.04 | **48.91** |
| | CIFAR10 | AWP | 81.01 | 55.36 | 50.12 | 81.61 | 55.05 | 49.85 |
| | | $OROAWP_{DA}$ | **81.12** | 55.89 | 50.49 | **81.63** | 55.32 | 50.19 |
| | | $OROAWP_{AS}$ | 78.68 | **56.52** | **50.75** | 79.06 | **55.70** | **50.59** |
| | CIFAR10 | MLCAT | 81.70 | 58.33 | 50.54 | 82.26 | 58.25 | 50.46 |
| | | $OROMLCAT_{DA}$ | **82.06** | 58.76 | 50.61 | **82.50** | 58.57 | 50.52 |
| | | $OROMLCAT_{AS}$ | 77.12 | **59.01** | **50.83** | 78.79 | **58.79** | **50.68** |
| Wide ResNet-34-10 | CIFAR10 | AT | **85.49** | 55.40 | 52.31 | **86.50** | 47.14 | 45.74 |
| | | $OROAT_{AS}$ | 82.64 | **59.07** | **53.04** | 82.71 | **49.68** | **46.59** |
| | CIFAR100 | AT | 60.90 | 31.35 | 27.42 | 59.07 | 26.03 | 24.39 |
| | | $OROAT_{AS}$ | 56.55 | **33.04** | **28.58** | 52.75 | **27.23** | **24.58** |
| | CIFAR10 | TRADES | **84.78** | 56.25 | 53.12 | **84.70** | 48.49 | 46.69 |
| | | $OROTRADES_{AS}$ | 83.36 | **57.07** | **53.79** | 84.64 | **49.48** | **47.25** |
| | CIFAR10 | AWP | **85.30** | 58.35 | 53.07 | **85.39** | 57.16 | 52.49 |
| | | $OROAWP_{AS}$ | 84.47 | **59.67** | **54.35** | 84.87 | **57.66** | **52.93** |
| | CIFAR10 | MLCAT | **86.72** | 62.63 | 54.73 | **87.32** | 61.91 | 54.61 |
| | | $OROMLCAT_{AS}$ | 85.41 | **63.60** | **55.25** | 84.74 | **62.47** | **54.88** |

include two typical methods for mitigating robust overfitting: AWP (Wu et al., 2020) and MLCAT (Yu et al., 2022b). For training, we followed the same optimization parameters as in Rice et al. (2020) for a fair comparison. In terms of evaluation, we utilized PGD-20 (Madry et al., 2018) and AutoAttack (AA) (Croce & Hein, 2020) as adversarial attack methods. The detailed descriptions of the experimental setup are in the Appendix E.

## 4.1 ROBUSTNESS EVALUATION

The evaluation results of $OROAT_{AS}$ and $OROAT_{DA}$ are summarized in Table 1. Here, "Best" refers to the highest achieved robustness during training, "Last" refers to the robustness of the checkpoint at the last epoch, and "Natural" denotes the accuracy on normal data. Note that we use the prefix "ORO" to denote the corresponding baselines that have integrated our proposed method. For example, if the attack strength strategy is applied to TRADES, we represent it as $OROTRADES_{AS}$. We observe that the proposed approaches significantly enhance adversarial robustness over standard AT, demonstrating the effectiveness of $OROAT_{AS}$ and $OROAT_{DA}$. Furthermore, this performance improvement is consistent across different datasets, network architectures, and adversarial training variants, indicating that our proposed methods reliably enhance adversarial robustness. Moreover, it is worth noting that both AWP and MLCAT have already effectively mitigated robust overfitting. The proposed approaches still contribute to a complementary improvement in adversarial robustness. The enhanced adversarial robustness on AWP and MLCAT by our proposed methods further highlights the significance of understanding the underlying mechanisms of robust overfitting.

## 4.2 ANALYSIS AND DISCUSSION

**Ablation analysis.** To analyze the role of the introduced attack strength component and data augmentation component in mitigating robust overfitting and enhancing adversarial robustness, we conducted an ablation study with standard AT using PreAct ResNet-18 on the CIFAR10 dataset. Specifically, we varied the perturbation budget in the attack strength component from 0/255 to 24/255 and

Table 2: Ablation study of OROAT$_{\text{AS}}$ and OROAT$_{\text{DA}}$ methods. The results were calculated as the average of three random trials.

| Method | Budget/Rate | Best | | | Last | | |
|---|---|---|---|---|---|---|---|
| | | Natural | PGD-20 | AA | Natural | PGD-20 | AA |
| OROAT$_{\text{AS}}$ | 0/255 | **84.57**±**0.23** | 50.75±0.10 | 45.17±0.06 | **86.71**±**0.38** | 41.30±0.14 | 36.61±0.13 |
| | 4/255 | 83.94±0.40 | 51.09±0.23 | 46.26±0.03 | 85.68±0.21 | 41.38±0.26 | 39.11±0.12 |
| | 8/255 | 81.92±0.46 | 51.96±0.14 | 47.74±0.12 | 83.87±0.36 | 43.56±0.06 | 41.42±0.02 |
| | 12/255 | 80.49±0.57 | 54.80±0.08 | 48.59±0.08 | 80.85±0.42 | 49.99±0.19 | 45.36±0.16 |
| | 16/255 | 77.48±0.36 | **56.35**±**0.20** | **49.11**±**0.14** | 76.84±0.28 | 53.20±0.16 | **46.24**±**0.06** |
| | 20/255 | 75.07±0.49 | 55.90±0.15 | 48.19±0.08 | 73.97±0.31 | **53.24**±**0.20** | 45.46 ±0.08 |
| | 24/255 | 74.24±0.21 | 54.71±0.06 | 46.86±0.05 | 72.71±0.50 | 52.54±0.10 | 44.63±0.03 |
| OROAT$_{\text{DA}}$ | 1.0 | 82.00±0.30 | 52.17±0.17 | 47.77±0.08 | 84.37±0.36 | 43.96±0.25 | 41.61±0.19 |
| | 0.8 | 82.42±0.38 | 52.26±0.24 | 48.08±0.15 | 84.61±0.47 | 47.76±0.12 | 43.78±0.02 |
| | 0.6 | 82.58±0.42 | 53.95±0.08 | **48.48**±**0.08** | **85.45**±**0.52** | 49.69±0.25 | **44.44**±**0.15** |
| | 0.4 | 83.06±0.20 | 55.46±0.15 | 46.98±0.06 | 84.89±0.22 | 51.03±0.08 | 44.18±0.02 |
| | 0.2 | **83.08**±**0.35** | **55.99**±**0.22** | 45.18±0.16 | 84.83±0.40 | **52.53**±**0.17** | 43.29±0.08 |

adjusted the threshold for small-loss adversarial data in the data augmentation component from 1.0 to 0.2. The results are summarized in Table 2. For robust overfitting, it was observed that as these components became more aggressive, such as when the attack strength component removes more non-effective features or when the data augmentation component further decreases the model's robustness on the training dataset, the degree of robust overfitting becomes milder. These results strongly support our analysis regarding the onset of robust overfitting. On the other hand, regarding the model's adversarial robustness, we observed a trend of initially increasing and then decreasing. The observed trend can be attributed to the effects of these introduced components. While effective in suppressing robust overfitting, these components also have a detrimental effect on the model's adversarial robustness. For example, the attack strength component will also eliminate some useful robust features, and the data augmentation component will degrade the model's defense against strong attacks. In the early stages, the advantage of these components in suppressing robust overfitting is predominant, leading to an overall improvement in the model's robustness. However, as these components become more aggressive, their disadvantages eventually outweigh the benefits of suppressing robust overfitting, resulting in a decrease in the model's robustness.

**Discussion.** The proposed approach introduces additional components into the adversarial training framework, thus increasing computational complexity. For the attack strength component, its computational cost depends on the perturbation budget; the larger the budget, the more additional attack iterations are required. Regarding the data augmentation component, the computational cost of this component is primarily influenced by the threshold set for small-loss adversarial data. When the threshold is low, a significant computational cost is needed to meet algorithm objectives due to the stochastic nature of data augmentation techniques. Due to its high computational cost, we restricted experiments involving the data augmentation component to the low-capacity PreAct ResNet-18, as shown in Table 1. While the proposed methods may not represent the optimal algorithm for addressing robust overfitting, especially in consideration of computational complexity and adversarial robustness, we want to emphasize that their design was intended to support our analysis of the onset of robust overfitting. Furthermore, their experimental results strongly validate our analysis and have demonstrated substantial improvements in adversarial robustness across a wide range of baselines. We hope that our understanding of the underlying mechanisms of robust overfitting will inspire future research to explore more efficient methods for handling robust overfitting.

## 5 CONCLUSION

In this work, we develop factor ablation adversarial training and identify that the contributing factors to robust overfitting originate from normal data. Furthermore, we analysis the onset of robust overfitting as a result of learning non-effective features and provide a comprehensive understanding of robust overfitting. To support our analysis, we design two orthogonal approaches: attack strength derived from feature elimination and data augmentation derived from robustness alignment. Extensive experiments validate our analysis and demonstrate the effectiveness of the proposed approaches in enhancing adversarial robustness across different adversarial training methods, network architectures, and benchmark datasets.

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

## A   ADDITIONAL EVIDENCE FOR FACTOR ABLATION ADVERSARIAL TRAINING

In this section, we present additional empirical evidence regarding factor ablation adversarial training across different datasets, network architectures, and adversarial training variants. We use the same experimental setup as described in Section 3.1, where we selectively remove specific ablation factors during training. Specifically, in the **data & perturbation** group, we remove both the normal data and the adversarial perturbations from the small-loss adversarial data. In the **perturbation** group, we solely remove the adversarial perturbations from the small-loss adversarial data. As shown in Figure 3, the **data & perturbation** group consistently exhibits a significant relief in robust overfitting, while the **perturbation** group demonstrates severe robust overfitting. These results strongly suggest that the contributing factors to robust overfitting originating from the normal data are generalizable across different settings in adversarial training.

## B   REVISITING EXISTING TECHNIQUES FOR MITIGATING ROBUST OVERFITTING

**Sample reweighting.** Sample reweighting is a common technique in AT used to mitigate robust overfitting. It assigns weighted values to each adversarial data point, differentiating the importance of various training data. We've observed that the current literature employs the sample reweighting technique in diverse ways. For example, Zhang et al. (2020) used sample reweighting to weaken the model's learning on small-loss adversarial data, while Yu et al. (2022b) used it to strengthen the model's learning on small-loss adversarial data. These two approaches utilize sample reweighting with entirely opposing objectives, yet both effectively alleviate robust overfitting. Our analysis can explain why both methods are effective in mitigating robust overfitting: the sample reweighting technique in Zhang et al. (2020) reduces the importance of small-loss adversarial data, essentially weakening the role of non-effective features learned on these data in model optimization, thus effectively mitigating robust overfitting. The sample reweighting technique in Yu et al. (2022b) increases the adversarial loss of small-loss adversarial data, essentially narrowing the model's robustness gap between the training and test sets. This reduces the generation of non-effective features and thereby effectively alleviates robust overfitting. In summary, one approach weakens the importance of non-effective features in model optimization, while the other decreases the generation of non-effective features. Although the objectives of these two methods are completely opposite, both lead to a reduction in the model's learning of non-effective features, and thus effectively alleviate robust overfitting.

**Additional training data.** Incorporating additional training data is a typical strategy to address robust overfitting. For instance, Carmon et al. (2019); Alayrac et al. (2019); Zhai et al. (2019) introduce more training data through semi-supervised learning to mitigate robust overfitting and enhance adversarial robustness in AT. However, it remains unclear how much extra training data is required to prevent robust overfitting (Gowal et al., 2020), and in some cases, additional training data may not necessarily alleviate robust overfitting (Chen et al., 2020a; Min et al., 2021). Our analysis offers intuitive explanations for these issues: as discussed in Section 3.2, robust overfitting arises from the model learning non-effective features, and a crucial condition for the generation of non-effective features is a substantial robustness gap between the training and test datasets. The strategy of additional training data can directly influences the model's robustness on the training data, making it an effective method to prevent robust overfitting. On the other hand, if the added training data fails to narrow the robustness gap between the training and test datasets or does not restrain the impact of non-effective features in model optimization, it will be ineffective against robust overfitting. In summary, for the technique of using additional training data, the quantity of extra training data is not the determining factor. Instead, its effectiveness hinges on whether these added training data can narrow the model's robustness gap between the training and test datasets or overwhelm the influence of non-effective features in model optimization.

**Data augmentation.** Data augmentation techniques involve applying random transformations to training data during the training process. It has been empirically shown to reduce overfitting in standard training. However, previous attempts (Gowal et al., 2020; Rebuffi et al., 2021) have shown that data augmentation doesn't provide much help for robust overfitting in AT. Later, some evidence suggests that data augmentation can be effective when combined with regularization (Tack

et al., 2022) or when used alone (Li & Spratling, 2023). Similarly, data augmentation also allows for direct adjustments to the model's robustness on the training dataset. Thus, it can serve as an effective approach to address robust overfitting. However, data augmentation techniques generally involve random image transformations and may not always achieve the desired effect. In particular, we utilize data augmentation techniques to design a method for achieving the alignment of the model's robustness between the training and test sets. We demonstrate that simple data augmentation methods with a targeted transformation objective can be significantly helpful in alleviating robust overfitting and enhancing adversarial robustness.

## C  ADDITIONAL EVIDENCE FOR OROAT$_\text{AS}$

In this section, we present additional results for OROAT$_\text{AS}$. We conduct OROAT$_\text{AS}$ experiments across different datasets, network architectures, and adversarial training variants. The results are summarized in Figure 4. Consistently, we observe a clear correlation between the applied attack strength and the extent of robust overfitting. The more non-effective features are eliminated, the milder the degree of robust overfitting. Furthermore, when the attack strength exceeds a certain threshold, the model exhibits almost no robust overfitting. These experimental results provide compelling evidence that it is these non-effective features that lead to robust overfitting.

## D  ADDITIONAL EVIDENCE FOR OROAT$_\text{DA}$

In this section, we present additional evidence supporting the effectiveness of OROAT$_\text{DA}$. We conducted OROAT$_\text{DA}$ experiments using different datasets, network architectures, and adversarial training variants. The summarized results are shown in Figure 5. Consistently, we observe a clear correlation between the proportion of small-loss data and the extent of robust overfitting. As the model's robustness on the training dataset decreases, the degree of robust overfitting becomes increasingly mild. These results fully validate our analysis of OROAT, demonstrating that the model's robustness gap promotes the generation of non-effective features.

## E  EXPERIMENTAL SETUP

Our project is implemented in the PyTorch framework on a server equipped with four GeForce GTX 3090 GPUs. The code and related models will be publicly released for verification and use. Our experiments employ the infinity norm as the adversarial perturbation constraint, and cover different benchmark datasets (CIFAR10 and CIFAR100 (Krizhevsky et al., 2009)), network architectures (PreAct ResNet-18 (He et al., 2016) and Wide ResNet-34-10 (Zagoruyko & Komodakis, 2016)), and adversarial training approaches (AT (Madry et al., 2018), TRADES (Zhang et al., 2019), AWP (Wu et al., 2020) and MLCAT (Yu et al., 2022b)).

We follow the training settings outlined in Rice et al. (2020), where the network is trained for 200 epochs using stochastic gradient descent (SGD) with momentum 0.9, weight decay of $5 \times 10^{-4}$, and an initial learning rate of 0.1. The learning rate is divided by 10 at the 100th and 150th epoch, respectively. Standard data augmentation techniques, including random cropping with 4 pixels of padding and random horizontal flips, are applied. For adversarial training, we use a 10-step PGD attack with a perturbation budget of $\epsilon = 8/255$ and step size of $\alpha = 2/255$, which is a standard setting in PGD-based adversarial training (Madry et al., 2018).

We evaluate model robustness under various criteria, including natural accuracy, 20-step PGD (PGD-20) (Madry et al., 2018), and AutoAttack (AA) (Croce & Hein, 2020). The attack step of the adversary in OROAT$_\text{AS}$ linearly increases with the perturbation budget, i.e., a 10-step PGD for $\epsilon = 8/255$ and a 20-step PGD for $\epsilon = 16/255$. The detailed hyperparameters are shown in Table 3. Other hyperparameters of the baselines are set as per their original papers.

## F  THE IMPACT OF DATA AUGMENTATION

The role of the data augmentation component in the OROAT$_\text{DA}$ method is to align the model's robustness between the training and test sets. In this section, we investigate the impact of the adopted

Table 3: The hyperparameter settings for OROAT$_{AS}$ and OROAT$_{DA}$.

| Network | Dataset | Method | Hyperparameter | | |
| --- | --- | --- | --- | --- | --- |
| | | | Small-loss Threshold t | Perturbation Budget $\epsilon_a$ | Small-loss Proportion p |
| PreAct ResNet-18 | CIFAR10 | OROAT$_{DA}$ | 1.7 | - | 0.6 |
| | | OROAT$_{AS}$ | 1.7 | 14/255 | - |
| | CIFAR100 | OROAT$_{DA}$ | 3.5 | - | 0.6 |
| | | OROAT$_{AS}$ | 3.5 | 15/255 | - |
| | CIFAR10 | OROTRADES$_{DA}$ | 1.9 | - | 0.8 |
| | | OROTRADES$_{AS}$ | 1.9 | 10/255 | - |
| | CIFAR10 | OROAWP$_{DA}$ | 0.8 | - | 0.8 |
| | | OROAWP$_{AS}$ | 0.8 | 12/255 | - |
| | CIFAR10 | OROMLCAT$_{DA}$ | 0.8 | - | 0.8 |
| | | OROMLCAT$_{AS}$ | 0.8 | 11/255 | - |
| Wide ResNet-34-10 | CIFAR10 | OROAT$_{AS}$ | 1.4 | 13/255 | - |
| | CIFAR100 | OROAT$_{AS}$ | 4.2 | 16/255 | - |
| | CIFAR10 | OROTRADES$_{AS}$ | 1.9 | 10/255 | - |
| | CIFAR10 | OROAWP$_{AS}$ | 1.1 | 10/255 | - |
| | CIFAR10 | OROMLCAT$_{AS}$ | 1.1 | 10/255 | - |

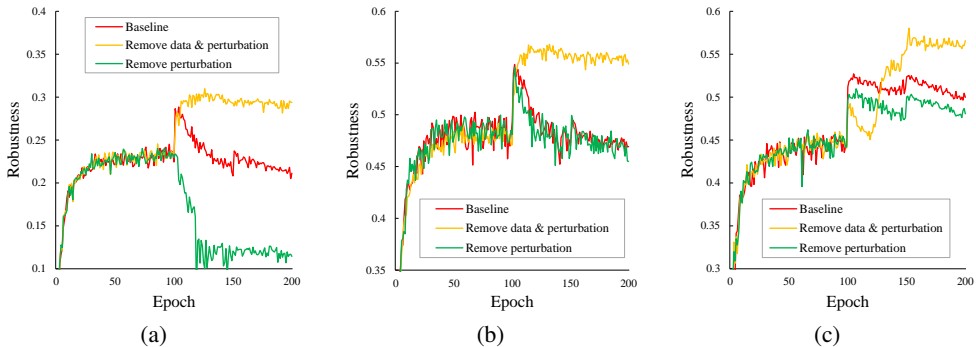

Figure 3: Experimental results for factor ablation adversarial training (a) on the CIFAR100 dataset using PreAct ResNet-18 with AT, (b) on the CIFAR10 dataset using Wide ResNet-34-10 with AT, and (c) on the CIFAR10 dataset using PreAct ResNet-18 with TRADES.

data augmentation methods on the experimental results. Specifically, we assessed three popular data augmentation methods - AutoAugment, RandAugment, and TrivialAugment - for the OROAT$_{DA}$ algorithm. For more detailed information on these data augmentation methods, please refer to Cubuk et al. (2019; 2020); Müller & Hutter (2021). The experimental results are summarized in Figure 6. We can observe that the stability of these data augmentation methods is comparatively lower than that of the AugMix method (Hendrycks et al., 2020). This is attributed to the fact that these transformations introduce more substantial alterations to the semantic content of the original image. Nonetheless, we observe that all data augmentation methods are capable of effectively mitigating robust overfitting by decreasing the proportion of small-loss adversarial data to a certain extent. This indicates that the proposed OROAT$_{DA}$ is generally effective regardless of the choice of data augmentation methods.

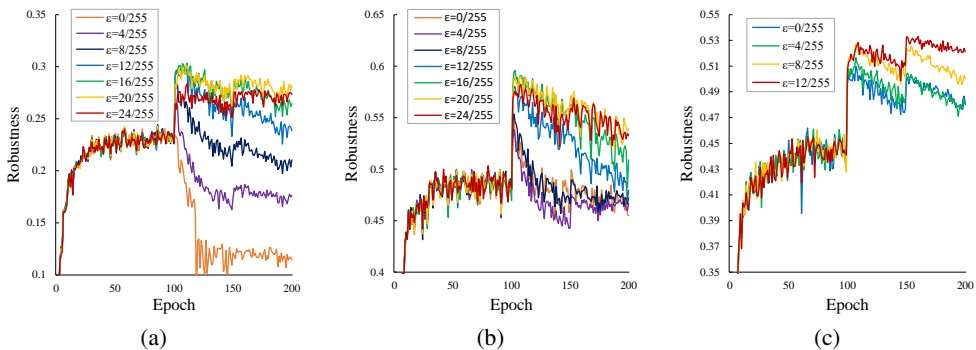

Figure 4: The test robustness of OROAT$_{AS}$ (a) on the CIFAR100 dataset using PreAct ResNet-18 with AT, (b) on the CIFAR10 dataset using Wide ResNet-34-10 with AT, and (c) on the CIFAR10 dataset using PreAct ResNet-18 with TRADES.

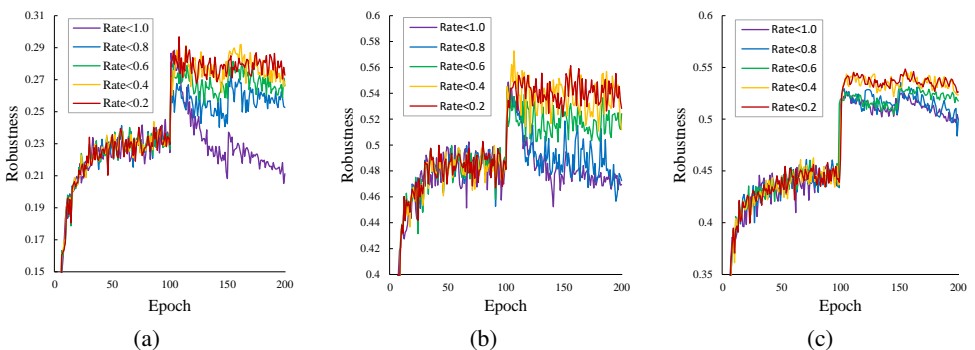

Figure 5: The test robustness of OROAT$_{DA}$ (a) on the CIFAR100 dataset using PreAct ResNet-18 with AT, (b) on the CIFAR10 dataset using Wide ResNet-34-10 with AT, and (c) on the CIFAR10 dataset using PreAct ResNet-18 with TRADES.

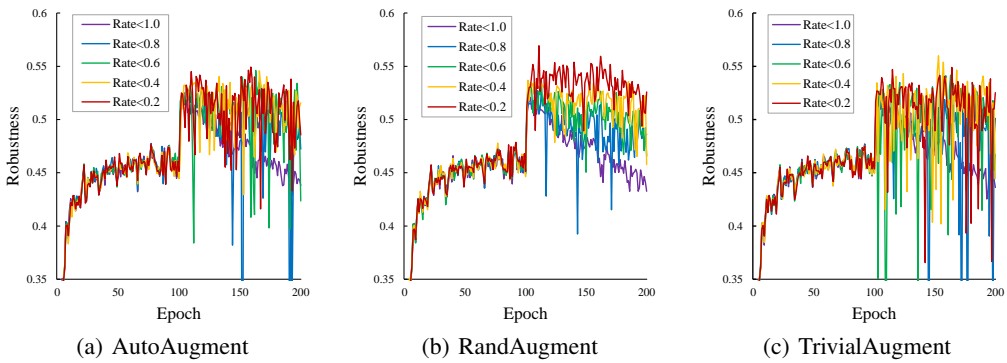

(a) AutoAugment      (b) RandAugment      (c) TrivialAugment

Figure 6: The test robustness of the OROAT$_{DA}$ method with three different data augmentation techniques: (a) AutoAugment; (b) RandAugment, and (c) TrivialAugment.

