# OpenReview forum: "On the Onset of Robust Overfitting in Adversarial Training"
_ICLR.cc/2024/Conference — Submitted to ICLR 2024_

### Official Review · Reviewer_BeRL · 2023-10-28

**Soundness:** 2 fair
**Presentation:** 1 poor
**Contribution:** 2 fair
**Rating:** 1
**Confidence:** 4

**Summary:**

This paper proposes a new method to mitigate overfitting in adversarial training. The proposed method changes the magnitude of adversarial attacks in adversarial training and applies AugMix to the small-loss examples within a minibatch if the proportion of the small-loss examples is below the specified threshold. Experiments demonstrate that the proposed method improves the performance of AT, AWP, TRADES, and MLCAT.

**Strengths:**

- This paper addresses an important problem: overfitting in adversarial training because adversarial training suffers from overfitting more than standard training.
- Experiments use AutoAttack, which is a de-fact standard evaluation method, and several baselines are used.
- Figure 1 shows somewhat interesting results. Comparing removing small-loss adversarial examples with removing only the perturbation of small-loss adversarial examples is an interesting investigation from a new aspect. However, this result does not lead well to the proposed method.

**Weaknesses:**

- There are a lot of undefined words: e.g., non-effective features and learning state. Since these words are frequently used in the analysis for developing the proposed method, readers could not understand why the proposed method is effective to mitigate overfitting.
Non-effective features and learning state should be defined by using equations and empirically evaluated or theoretically evaluated in the existence of adversarial training.
- Most parts of claim do not have objective evidence. For scientific articles, most of claims should be supported by the evidence. For example, the following states do not supported by the experimental or theoretical results:
_"In the initial stages of adversarial training, due to the similarity in the model’s learning states between the training and test datasets, the boundary between the robust and non-robust features doesn’t significantly differ between the training and test sets. "_
_"However, the improvement in the model’s learning state on the test dataset is relatively limited, far from matching the model’s learning state on the training dataset"_
_" As a result, the boundary between the robust and non-robust features becomes progressively more distinct between the training and test sets."_
_"adversarial data with small loss indicates that the model’s learning state on these data is excellent, maintaining a substantial gap compared to the learning state on the test set."_
I suggest you to provide more empirical results or theoretical results that support your claims and the effectiveness of the proposed method.
- The proposed method is not clearly written, and its explanation does not have equations or pseudo codes.
Readers cannot reproduce the results.
I could not understand how to control the attack strength in the proposed method and how to use data augmentation.
What value of the loss do you call small loss for small-loss adversarial data?
What is the specified threshold for the proposed method?
Regarding changing adversarial budgets in adversarial training, [a] might be related work, which schedules adversarial budgets for considering loss landscapes.

[a] Liu, Chen, et al. "On the loss landscape of adversarial training: Identifying challenges and how to overcome them." Advances in Neural Information Processing Systems 33 (2020): 21476-21487.

**Questions:**

- Do you have any evidence that supports your claims as witten in Weakness?

---

> ### Author Response · Authors · 2023-11-17
> **Response to Reviewer BeRL**
>
> We thank the reviewer for the insightful comments.
>
> **Q1:** Undefined words and lack evidence to support the claims.\
> **A1:** The non-effective features denote the robust features in the training set that lack generalization. The term "learning state gap" between the training and test sets has been revised to "robustness gap" in our updated manuscript. This change has been made consistently throughout the paper. Additionally, we employ quantitative metrics such as adversarial loss and robust accuracy to measure the model's robustness, providing evidential support for the existence of a robustness gap between the training and test datasets, as illustrated in Figure 1(b). Regarding the generation of non-effective features, our analysis is as follows: during the adversarial training process, the model's adversarial robustness on the training set exceeds that on the test set, as evidenced by the adversarial loss and robust accuracy observed in Figure 1(b). This results in a robustness gap between the training and test data. Considering that adversarial perturbations are generated on-the-fly and adaptively adjusted based on the model's robustness. The robustness gap between the training and test data leads to distinct adversarial perturbation on the robust features in these datasets. The varying degree of adversarial perturbation on the robust features in the training and test data amplify the distribution differences between these two datasets, thereby degrading the generalization of robust features on the training set and facilitating the generation of non-effective features. The increasing non-effective features further exacerbates the robustness gap between training and test data, forming a vicious cycle. These analyses are detailed in our updated manuscript presented in Section 1, Section 3, and Figure 2. We appreciate the valuable suggestion to refine our analysis, and we believe this adjustment enhances the clarity and rigour of our research.
>
> **Q2:** The pseudocodes of the proposed methods.\
> **A2:** In the revised manuscript, we have incorporated the pseudocodes of the proposed methods in Algorithm 1 and Algorithm 2. We sincerely appreciate the valuable suggestion to refine our method, and we believe this adjustment enhances the logical flow of ideas.
>
> **Q3:** The hyperparameter settings of the proposed methods.\
> **A3:** In the revised manuscript, we have included detailed experimental hyperparameters in Table 3. We sincerely appreciate the valuable suggestion to refine our experiment, and we believe this adjustment enhances the clarity and rigor of our research.
>
> **Q4:** Related work.\
> **A4:** Thank you for pointing out this related work (Liu et al., PAS), and we have cited it in the revised manuscript. Similar to our attack strength approach, PAS also adjusts the adversarial budgets. However, it is important to note that there are some key differences:
> - **Different research problems.** PAS investigates the optimization challenges in adversarial training, specifically tackling the increased curvature and gradient scattering caused by the large adversarial budget. Instead, our focus is on the phenomenon of robust overfitting within adversarial training, specifically analyzing the onset of robust overfitting.
> - **Different perspectives.** The main focus/contribution of our work lies in understanding the generation of non-effective features. The attack strength method is designed based on our understanding to specifically eliminate non-effective features. Instead, the design of PAS is motivated directly by the influence of the perturbation budget on loss landscape. Therefore, the two works clearly have different motivations and design principles.
> - **Different solutions.** The two are also different in the strategies for adjusting perturbation budgets. PAS employs a periodic budget scheduling scheme where the perturbation budget dynamically changes along training and is applied to all samples. Instead, ours is much simpler and easy to use: the modified perturbation budget is constant and is applied solely to the small-loss adversarial samples.
>
> Thus, according to these key differences, our method is quite different from PAS and provides new perspectives and solutions for understanding and alleviating robust overfitting.
>
>
> Thank you once again for your valuable comments, which have helped us improve the clarity and rigour of our research. Hope our elaborations and the revisions made in the new manuscript could address your concerns. Please let us know if there is more to clarify.

---

> ### Author Response · Authors · 2023-11-20
> **Could you please have a look at our rebuttal?**
>
> Dear Reviewer BeRL, thanks for your time reviewing our paper. We have meticulously prepared a detailed response addressing the concerns you raised. Could you please have a look to see if there are further questions? Your invaluable input is greatly appreciated. Thank you once again, and we hope you have a wonderful day!

---

> > ### Comment · Reviewer_BeRL · 2023-11-21
> > **Thank you for the feedback!**
> >
> > I have read the author feedback. I am happy to see that pseudo codes clarify the proposed method, and some words are explained. However, the new results and analyses lack novelty, and there are still several undefined words and claims not supported by theory or empirical results. So, I keep my score and vote to reject.
> >
> >
> > > Additionally, we employ quantitative metrics such as adversarial loss and robust accuracy to measure the model's robustness, providing evidential support for the existence of a robustness gap between the training and test datasets, as illustrated in Figure 1(b). Regarding the generation of non-effective features, our analysis is as follows:
> >
> > Fig 1(b) only shows overfitting in adversarial training, which is well known e.g., (Rice et al., 2020). Then, the analyses derived from the results are important. However, the idea that the non-effective features increase is too crude. What do you define non-effective features as? If it is the feature vector of the model, which layer is it? And what features are non-effective? How was the existence of non-effective features confirmed? If non-effective features are just features that do not classify correctly and are confirmed by Fig. 1(b), this analysis provides no new information or insights at all.
> >
> > The unclearness and confusion of this analysis also affect the importance of the proposed method.  This is because you claim that (Liu et al., PAS) and the proposed method differ in terms of Different research problems and Different perspectives. If the analysis is not an important result, then the proposed method is not likely to yield interesting results either.
> >
> > I suggest that you first present the hypothesis about non-effective features with its definition, and design the experiments that can evaluate whether the hypothesis is true or not. The analysis should build on existing research and provide new findings. The proposed method should be explained based on the findings.

---

> > > ### Author Response · Authors · 2023-11-21
> > > **Further Response to Reviewer BeRL**
> > >
> > > Thanks for your reply! We will address your further questions below.
> > >
> > > >What do you define non-effective features as? If it is the feature vector of the model, which layer is it? And what features are non-effective?
> > >
> > > In the rebuttal, we have explained that “the non-effective features denote the robust features in the training set that lack generalization”. We have also provided a clear definition of non-effective features in Section 1 and Section 3: “certain non-generalizable robust features emerge in the training set, which we denote as non-effective features” and “Some robust features in the training set may lack generalization. We refer to these features as non-effective features.”
> > >
> > > >How was the existence of non-effective features confirmed?
> > >
> > > We conduct a series of rigorously factor ablation experiments following the principles of the controlled variable method, inferring that the factors inducing robust overfitting originate from the normal data. Furthermore, we design the OROAT$_\mathrm{AS}$ method to eliminate certain features in the training set, effectively alleviating robust overfitting and confirming the existence of non-effective features.
> > >
> > > >If non-effective features are just features that do not classify correctly and are confirmed by Fig. 1(b), this analysis provides no new information or insights at all.
> > >
> > > In the rebuttal and the updated manuscript, we explicitly state that Figure 1(b) is intended to support the robustness gap between the training and test datasets. Regarding how the robustness gap between the training and test datasets promotes the generation of non-effective features: adversarial perturbations are generated on-the-fly and adaptively adjusted based on the model's robustness. The robustness gap between the training and test data leads to distinct adversarial perturbation on the robust features in these datasets. The varying degree of adversarial perturbation on the robust features in the training and test data amplify the distribution differences between these two datasets, thereby degrading the generalization of robust features on the training set and facilitating the generation of non-effective features.
> > >
> > > >The new results and analyses lack novelty.
> > >
> > > If there is other related work that also explains robust overfitting from the generation of non-effective features like ours, please let us know.
> > >
> > > Hope the clarification above could address your concerns. Please let us know if there is more to clarify.

---

> > > > ### Author Response · Authors · 2023-11-21
> > > > **Would you consider reevaluating the paper?**
> > > >
> > > > We truly appreciate your comments and suggestions. We have provided extensive clarifications on the definition, existence, and generation of non-effective features. Considering that many of your previous concerns appear to have been resolved, could you kindly reconsider the evaluation of our paper? Your feedback is invaluable to us.

---

> ### Author Response · Authors · 2023-11-21
> **Could you please have a look to see if our response has resolved your concerns?**
>
> Dear Reviewer BeRL, thanks for your time reviewing our paper. We have prepared a detailed response addressing the further concerns you raised. Could you please have a look to see if our response has resolved your concerns? Your invaluable input is greatly appreciated. Once again, thank you, and we wish you a wonderful day!

---

> ### Comment · Reviewer_BeRL · 2023-11-22
> **Thank you for the further response**
>
> > We have also provided a clear definition of non-effective features in Section 1 and Section 3: “certain non-generalizable robust features emerge in the training set, which we denote as non-effective features” and “Some robust features in the training set may lack generalization. We refer to these features as non-effective features.”
>
> Let $f$ and $h$ be a classifier and feature extractor, and let $f\circ h(x)$ be the model. Features for training set become {$h(x)|x\in D_{train}$}. Then, what is robust features in the training set that may lack generalization? Generalization error is generally defined as the average of errors over test set $D_{test}$ as $E_{x\in D{test}} [R (f\circ h(x))]$, and thus, I cannot understand relationship between generalization and robust features in the training set $D_{train}$.
>
> This is just one example; there are too many similarly unclear or messy arguments. Further replies will not change my score as there are many major issues with the paper.

---

> > ### Author Response · Authors · 2023-11-22
> > **Thanks for your reply and further questions**
> >
> > Thanks for your reply! We will address your further questions below.
> >
> > > I cannot understand relationship between generalization and robust features in the training set.
> >
> > Sorry for the confusion! We now get your point. Let me elaborate on the concept of non-effective features by an intuitive example. The objective function of adversarial training is $\min_\theta \frac{1}{n}\sum_{i=1}^{n} \max_{\delta_i \in \Delta} \ell(f_\theta(x_{i }+\delta_i),y_{i})$. This is a min-max optimization problem, where the inner maximization process generates adversarial perturbations on-the-fly that maximizes the classification loss. According to the inner maximization process, we can regard the generated adversarial perturbation as a counteract to the highly relevant features in the sample. For example, for a model that is not robust, most of these highly relevant features may be non-robust features, so the generated adversarial perturbations mainly act on these non-robust features. For a model with good robustness, these highly relevant features may become robust features, so the generated adversarial perturbations mainly focus on these robust features. Then for models with robustness gaps, their perturbation strength for robust features is also different. It is these differences in perturbation strength on features that promote the generation of non-effective features. In short, adversarial perturbations are generated on-the-fly and adaptively adjusted based on the model’s robustness. The robustness gap between the training and test data leads to distinct adversarial perturbation on the robust features in these datasets. The varying degree of adversarial perturbation on the robust features in the training and test data amplify the distribution differences between these two datasets, thereby degrading the generalization of robust features on the training set and facilitating the generation of non-effective features. The increasing non-effective features further exacerbates the robustness gap between training and test data, forming a vicious cycle.
> >
> > > This is just one example; there are too many similarly unclear or messy arguments. Further replies will not change my score as there are many major issues with the paper.
> >
> > It's okay further replies will not change your score and I will not ask you to change your score. But during the reviewer-author discussion, it is my responsibility to answer the reviewer's questions and clarify the misunderstandings between reviewers and the authors. So I hope you can provide me with all unclear or messy arguments and I will do my best to clarify unnecessary misunderstandings between us. Then I also hope that you can clearly point out what major issues exist in our work, I will be very grateful!

---

### Official Review · Reviewer_6tzX · 2023-10-28

**Soundness:** 2 fair
**Presentation:** 2 fair
**Contribution:** 2 fair
**Rating:** 3
**Confidence:** 4

**Summary:**

Summary:
This study is dedicated to investigating the fundamental mechanism of robust overfitting in adversarial training. First, the robust overfitting is attributed to the non-effective features that hinder the model from learning generalization ability. Then, the study proposes OROAT with attack strength and data augmentation to alleviate learning on non-effective features. Experiments validate the robustness of OROAT across several adversarial training methods to counter different attacks.

**Strengths:**

pros:
1. Provide an innovative view to separate adversarial training data into normal and small-loss adversarial data. And the method of composition that first dives into the problem and then proposes a solution is appealing.
2. Introduce a plug-and-play method that is experimented with many adversarial attack and training methods.

**Weaknesses:**

cons:
1. Chapter 3 lacks experimental results to support analysis. For example, the study mentions the gap between training and test learning state several times, which would be better accompanied by a figure illustrating the difference of robust overfitting between training and test data.
2. The writing of analysis and method is too lengthy, whereas the experiment part is too condensed. These sections have to be reorganized to alleviate the reading burden.
3. Experiments are solely conducted on CIFAR10 and CIFAR100. Larger datasets like ImageNet should be explored.
4. The ablation studies show that the effect of OROAT turns negative towards adversarial robustness and possible reasons. How to avoid this situation needs explanation. Additionally, Table2 needs to highlight the results that perform best or correspond to the argument in texts.
5. Lack comparisons with existing mitigations of robust overfitting. The authors has included various previous works in the related works/revisiting section, which is good. However, there is no empirical comparison.

**Questions:**

Refer to the weakness section.

---

> ### Author Response · Authors · 2023-11-17
> **Response to Reviewer 6tzX**
>
> We thank the reviewer for the insightful comments.
>
> **Q1:** Lack experimental results to support analysis.\
> **A1:** The term "learning state gap" between the training and test sets has been revised to "robustness gap" in our updated manuscript. This change has been made consistently throughout the paper. Additionally, we employ quantitative metrics such as adversarial loss and robust accuracy to measure the model's robustness, providing evidential support for the existence of a robustness gap between the training and test datasets, as illustrated in Figure 1(b). Regarding the generation of non-effective features, our analysis is as follows: during the adversarial training process, the model's adversarial robustness on the training set exceeds that on the test set, as evidenced by the adversarial loss and robust accuracy observed in Figure 1(b). This results in a robustness gap between the training and test data. Considering that adversarial perturbations are generated on-the-fly and adaptively adjusted based on the model's robustness. The robustness gap between the training and test data leads to distinct adversarial perturbation on the robust features in these datasets. The varying degree of adversarial perturbation on the robust features in the training and test data amplify the distribution differences between these two datasets, thereby degrading the generalization of robust features on the training set and facilitating the generation of non-effective features. The increasing non-effective features further exacerbates the robustness gap between training and test data, forming a vicious cycle. These analyses are detailed in our updated manuscript presented in Section 1, Section 3, and Figure 2. We appreciate the valuable suggestion to refine our analysis, and we believe this adjustment enhances the clarity and rigour of our research.
>
> **Q2:** Reorganize the manuscript.\
> **A2:** In the revised manuscript, we have relocated the analysis of existing techniques for mitigating robust overfitting to the appendix to alleviate the reading burden. Additionally, we've incorporated the pseudocodes of the proposed methods in Algorithm 1 and Algorithm 2 to enhance the logical flow of ideas. Thank you for bringing this to our attention.
>
> **Q3:** Larger datasets like ImageNet.\
> **A3:** Due to constraints on computational resources, training adversarially robust models on a large-scale dataset like ImageNet is currently beyond our capacity. However, to the best of my knowledge, the phenomenon of robust overfitting is primarily investigated on standard benchmark datasets such as CIFAR10.
>
> **Q4:** Highlight the results in Table 2.\
> **A4:** In the revised manuscript, we have emphasized the best results in Table 2. Thank you for highlighting this matter.
>
> **Q5:** Negative effect of OROAT methods.\
> **A5:** The proposed methods are specifically designed to validate the underlying mechanisms of robust overfitting, as detailed in Algorithm 1 and Algorithm 2. These methods currently involve adjusting adversarial training at the sample level, which introduce some negative impact on the model's adversarial robustness. We think that by refining adversarial training at the feature level, it should be possible to alleviate these drawbacks. This is an intriguing problem, and we appreciate your attention to this issue.
>
> **Q6:** Lack empirical comparison.\
> **A6:** Indeed, we have conducted experiments on two advanced methods for suppressing robust overfitting, namely AWP and MLCAT. It’s noteworthy that both AWP and MLCAT have already effectively mitigated robust overfitting. The proposed approaches still contribute to a complementary improvement in adversarial robustness. The enhanced adversarial robustness on AWP and MLCAT by our proposed methods further highlights the significance of understanding the underlying mechanisms of robust overfitting.
>
> Thank you once again for your valuable comments, which have helped us improve the clarity and rigour of our research. Hope our elaborations and the revisions made in the new manuscript could address your concerns. Please let us know if there is more to clarify.

---

> > ### Author Response · Authors · 2023-11-23
> > **Could you please have a look at our rebuttal?**
> >
> > Dear Reviewer 6tzX, thanks for your time reviewing our paper. We have meticulously prepared a detailed response and updated manuscript addressing the concerns you raised. Could you please have a look to see if our response has resolved your concerns? Your invaluable input is greatly appreciated. Thank you once again, and we hope you have a wonderful day!

---

> ### Author Response · Authors · 2023-11-20
> **Could you please have a look at our rebuttal?**
>
> Dear Reviewer 6tzX, thanks for your time reviewing our paper. We have meticulously prepared a detailed response addressing the concerns you raised. Could you please have a look to see if there are further questions? Your invaluable input is greatly appreciated. Thank you once again, and we hope you have a wonderful day!

---

> ### Author Response · Authors · 2023-11-21
> **Could you please have a look at our rebuttal?**
>
> Dear Reviewer 6tzX, thanks for your time reviewing our paper. We have meticulously prepared a detailed response and updated manuscript addressing the concerns you raised. Could you please have a look to see if our response has resolved your concerns? Your invaluable input is greatly appreciated. Once again, thank you, and we wish you a wonderful day!

---

### Official Review · Reviewer_sV3q · 2023-10-30

**Soundness:** 2 fair
**Presentation:** 2 fair
**Contribution:** 2 fair
**Rating:** 3
**Confidence:** 4

**Summary:**

In this paper, robust overfitting, an interesting and important phenomena in adversarial training, is investigated. The main conclusion is robust overfitting is a result of learning non-effective features, which also leads to new enhancement method.

**Strengths:**

Since robust-overfitting is a specific phenomena in adversarial training. Indeed, it should consider the difference of natural example and adversarial perturbation, as the authors did. To observe the difference, they design good ablation experiments, which indeed could bring new thoughts.

**Weaknesses:**

- the main conclusion that overfitting is because of learning non-effective features is too trivial. It may be true for any type of overfitting but specifically suitable for adversarial training.

- as I said, it is good to design interesting experiments to find something. But still it is better to also include theoretical discussions, especially on the specific properties for adversarial training.

- the methods derived from the main conclusion is not interesting. Data augmentation is almost the most natural way to suppress overfitting. Attack strength adjustment is also common for adversarial training. For example. PGD-based on AT can be regarded as adaptive attack adjustment.

- It is OK if the authors choose to evaluate the proposed method numerically. However, the experiments should be enhanced largely. The performance should be compared not only to vanilla adversarial training but also other robust overfitting suppression method. Notice that the training time should be reported.

**Questions:**

please see the weakness.

---

> ### Author Response · Authors · 2023-11-17
> **Response to Reviewer sV3q**
>
> We thank the reviewer for the insightful comments.
>
> **Q1:** The conclusion is too trivial.\
> **A1:** It's worth noting that the main focus/contribution of our work lies in understanding the onset of robust overfitting through a detailed analysis of the generation of non-effective features, rather than drawing the conclusion that non-effective features cause robust overfitting. Regarding the generation of non-effective features, our analysis is as follows: during the adversarial training process, the model's adversarial robustness on the training set exceeds that on the test set, as evidenced by the adversarial loss and robust accuracy observed in Figure 1(b). This results in a robustness gap between the training and test data. Considering that adversarial perturbations are generated on-the-fly and adaptively adjusted based on the model's robustness. The robustness gap between the training and test data leads to distinct adversarial perturbation on the robust features in these datasets. The varying degree of adversarial perturbation on the robust features in the training and test data amplify the distribution differences between these two datasets, thereby degrading the generalization of robust features on the training set and facilitating the generation of non-effective features. The increasing non-effective features further exacerbates the robustness gap between training and test data, forming a vicious cycle. These analyses are detailed in our updated manuscript presented in Section 1, Section 3, and Figure 2. We appreciate the valuable suggestion to refine our analysis, and we believe this adjustment enhances the clarity and rigour of our research.
>
> **Q2:** Discussion on the specific properties for adversarial training.\
> **A2:** In Section 3, we humbly offer the discussions on the generation of non-effective features during the adversarial training process: during adversarial training, adversarial perturbations are generated on-the-fly and adaptively adjusted based on the model’s robustness. In the initial stages of adversarial training, due to the similarity in the model's robustness between the training and test datasets, the generated adversarial perturbations on the robust features are relatively close in these datasets. Thus, most robust features in the training set can still exhibit good generalization, thereby enhancing the model's adversarial robustness on the test set. However, as training progresses, the model's robustness in the training dataset increases significantly faster than its robustness on the test dataset, as evidenced in Figure 1(b) by the adversarial loss and robustness observed on both datasets. This leads to a widening robustness gap between the training and test datasets. Consequently, the generated adversarial perturbation on the robust features becomes progressively more distinct in these datasets, which degrades the generalization of robust features on the training set and facilitates the generation of non-effective features, causing the model to learn an increasing number of non-effective features. As non-effective features proliferate, the robustness gap between the training and test sets continues to grow, forming a vicious cycle, as illustrated in Figure 2. Once the model's optimization is governed by these non-effective features, the model's adversarial robustness on the test dataset will continue to decline. This, in turn, gives rise to the phenomenon of robust overfitting.
>
> **Q3:** The methods are not interesting.\
> **A3:** We regret that the proposed methods did not capture your interest. In fact, our proposed methods are specifically crafted to validate the underlying mechanisms of robust overfitting, as outlined in Algorithm 1 and Algorithm 2. It is worth noting that, while data augmentation is almost the most natural way to suppress overfitting, applying the same data augmentation techniques in adversarial training, following the standard practice, proves ineffective in mitigating robust overfitting. Similarly, although attack strength adjustment is common for adversarial training, employing the same attack strength adjustment in accordance with conventional manner does not alleviate robust overfitting. In summary, while our methods may seem relatively simple and common, they are purposefully designed to validate our analysis of robust overfitting. Furthermore, they consistently enhance adversarial robustness across a wide range of experimental settings, demonstrating their efficacy.

---

> ### Author Response · Authors · 2023-11-17
> **Response to Reviewer sV3q**
>
> **Q4:** Experimental comparison.\
> **A4:** Indeed, we have conducted experiments on two advanced methods for suppressing robust overfitting, namely AWP and MLCAT. It’s noteworthy that both AWP and MLCAT have already effectively mitigated robust overfitting. The proposed approaches still contribute to a complementary improvement in adversarial robustness. The enhanced adversarial robustness on AWP and MLCAT by our proposed methods further highlights the significance of understanding the underlying mechanisms of robust overfitting.
>
> Thank you once again for your valuable comments, which have helped us improve the clarity and rigour of our research. Hope our elaborations and the revisions made in the new manuscript could address your concerns. Please let us know if there is more to clarify.

---

> ### Author Response · Authors · 2023-11-20
> **Could you please have a look at our rebuttal?**
>
> Dear Reviewer sV3q, thanks for your time reviewing our paper. We have meticulously prepared a detailed response addressing the concerns you raised. Could you please have a look to see if there are further questions? Your invaluable input is greatly appreciated. Thank you once again, and we hope you have a wonderful day!

---

> > ### Comment · Reviewer_sV3q · 2023-11-21
> > **thanks for the explanation**
> >
> > Thanks for the reply. But I still think the idea is too intuitive and lacks of theoretical discussion specifically for adversarial training. The experiments are also not very strong, especially on the comparison with SOTA methods. So I would like to keep my score. Hope further insightful study can make this work more interesting.

---

> > > ### Author Response · Authors · 2023-11-21
> > > **Further Response to Reviewer sV3q**
> > >
> > > Thanks for your reply! We will address your remaining concerns as follows.
> > >
> > > >The idea is too intuitive.
> > >
> > > In our rebuttal, we have explained that the main focus/contribution of our work lies in the analysis of the generation of non-effective features. For the generation of non-effective features, we have also provided detailed explanations within the AT framework. We do not fully understand your concerns that the idea is too intuitive and respectfully hope that you could provide concrete explanations for this concern.
> > >
> > > >Lacks of theoretical discussion.
> > >
> > > As for the concern about the lack of theoretical discussion, we note that this is mainly because robust overfitting is a complex phenomenon of DNNs and it is hard to be realized with theoretically tractable toy models. As far as we know, existing explanations of robust overfitting mostly rely on empirical justifications and do not have theoretical proof. In view of this situation, we also designed extensive factor ablation experiments and verification methods and they all agree well with our analysis. Thus, we believe that our analysis from the generation of non-effective features can further the understanding of robust overfitting.
> > >
> > > >The experiments are also not very strong.
> > >
> > > In our rebuttal, we have explained that the proposed methods are specifically crafted to validate our analysis of robust overfitting. For robustness performance, they consistently enhance adversarial robustness across a wide range of experimental settings, demonstrating their effectiveness. Besides, they can also contribute to the complementary robustness improvement on the comparison with SOTA methods, highlighting the significance of our analysis to robust overfitting.
> > >
> > > We are looking forward to more concrete explanations of the above limitations you mentioned.

---

> > > ### Author Response · Authors · 2023-11-23
> > > **Could you please have a look to see if our response has resolved your concerns?**
> > >
> > > Dear Reviewer sV3q, thanks for your time reviewing our paper. We have prepared a detailed response addressing the further concerns you raised. Could you please have a look to see if our response has resolved your concerns? Your invaluable input is greatly appreciated. Thank you once again, and we hope you have a wonderful day!

---

> ### Author Response · Authors · 2023-11-21
> **Could you please have a look to see if our response has resolved your concerns?**
>
> Dear Reviewer sV3q, thanks for your time reviewing our paper. We have prepared a detailed response addressing the further concerns you raised. Could you please have a look to see if our response has resolved your concerns? Your invaluable input is greatly appreciated. Once again, thank you, and we wish you a wonderful day!

---

### Official Review · Reviewer_H4Yh · 2023-11-08

**Soundness:** 2 fair
**Presentation:** 2 fair
**Contribution:** 2 fair
**Rating:** 3
**Confidence:** 4

**Summary:**

The paper addresses the problem of robust overfitting in adversarial training and seeks to understand and mitigate it through empirical analysis. They conduct factor ablation experiments in adversarial training and conclude that robust overfitting stems from the normal data. They explain the onset of robust overfitting is due to the learning of non-effective features during adversarial training, and revisit different techniques for mitigating robust overfitting from this perspective. Based on these insights, they propose two methods based on the attack strength and data augmentation to suppress the learning of non-effective features, and thereby reduce robust overfitting.

**Strengths:**

- The paper addresses robust overfitting in adversarial training, which is an important problem and not fully understood.

- The discussion of the phenomenon of robust overfitting leading to section 3.2 is clear and well motivated.

**Weaknesses:**

The central discussions of the paper are vague and mainly intuitive in nature. There are no concrete equations or an algorithm for the proposed methods based on attack strength and data augmentation. Furthermore, there is no analysis to support the claims about robust overfitting.

The discussions in section 3.2 and 3.3, which form the crux of the paper, are not clear. For instance, in the following statements (in Section 3.2.1, page 5), what is meant by the similarity of the model’s learning states on the training vs test data? The paper should explain this in a more principled, mathematical way.

> “In the initial stages of adversarial training, due to the similarity in the model’s learning states between the training and test datasets, the boundary between the robust and non-robust features doesn’t significantly differ between the training and test sets.”

> “Adversarial data with small loss indicates that the model’s learning state on these data is excellent, maintaining a substantial gap compared to the learning state on the test set.”

**Questions:**

1. In Eqn (4), please clarify that the max is over the perturbation $\delta_i \in \Delta$, and that $x^\prime_i = x_i + \delta_i$.

2. For the factor ablation experiments in Section 3.1 and Figure 1, are the results averaged over a few trials to account for randomness?

3. Can you concretely define "effective features" and the idea of "similarity of a model's learning states between the training and test data"?

---

> ### Author Response · Authors · 2023-11-17
> **Response to Reviewer H4Yh**
>
> We thank the reviewer for the insightful comments.
>
> **Q1:** Factor ablation experiments over a few trials.\
> **A1:** The results of factor ablation experiments in Figure 1 show the outcomes of a single run. However, it is worth noting that we have actually presented the results of these factor ablation experiments across different datasets, network architectures, and adversarial training variants, as illustrated in Figure 3. This broader presentation suggests that the observed phenomenon is a general finding in adversarial training.
>
> **Q2:** No algorithm for the proposed methods.\
> **A2:** In the revised manuscript, we have incorporated the pseudocodes of the proposed methods in Algorithm 1 and Algorithm 2. We sincerely appreciate the valuable suggestion to refine our method, and we believe this adjustment enhances the clarity of our research.
>
> **Q3:** No analysis to support the claims about robust overfitting.\
> **A3:** The term "learning state gap" between the training and test sets has been revised to "robustness gap" in our updated manuscript. This change has been made consistently throughout the paper. Additionally, we employ quantitative metrics such as adversarial loss and robust accuracy to measure the model's robustness, providing evidential support for the existence of a robustness gap between the training and test datasets, as illustrated in Figure 1(b). Regarding the process of how the robustness gap induces robust overfitting, our analysis is as follows: during the adversarial training process, the model's adversarial robustness on the training set exceeds that on the test set, as evidenced by the adversarial loss and robust accuracy observed in Figure 1(b). This results in a robustness gap between the training and test data. Considering that adversarial perturbations are generated on-the-fly and adaptively adjusted based on the model's robustness. The robustness gap between the training and test data leads to distinct adversarial perturbation on the robust features in these datasets. The varying degree of adversarial perturbation on the robust features in the training and test data amplify the distribution differences between these two datasets, thereby degrading the generalization of robust features on the training set and facilitating the generation of non-effective features. The increasing non-effective features further exacerbates the robustness gap between training and test data, forming a vicious cycle. These analyses are detailed in our updated manuscript presented in Section 1, Section 3, and Figure 2. We appreciate the valuable suggestion to refine our analysis, and we believe this adjustment enhances the clarity and rigour of our research.
>
> **Q4:** The definition of "effective features".\
> **A4:** The term "effective features" has been deprecated in our revised manuscript. We now explicitly denote them as robust features in the training dataset with generalization. For instance, "In the initial stages of adversarial training, due to the similarity in the model's robustness between the training and test datasets, the generated adversarial perturbation on the robust features is relatively close in these datasets. Thus, most robust features in the training set can still exhibit good generalization, thereby enhancing the model's adversarial robustness on the test set." We appreciate the feedback and have made the necessary adjustments to improve clarity.
>
> **Q5:** The error in Eqn(4).\
> **A5:** We have rectified the error in Eqn (4) in the revised manuscript. Thank you for bringing it to our attention.
>
> Thank you once again for your valuable comments, which have helped us improve the clarity and rigour of our research. Hope our elaborations and the revisions made in the new manuscript could address your concerns. Please let us know if there is more to clarify.

---

> ### Author Response · Authors · 2023-11-20
> **Could you please have a look at our rebuttal?**
>
> Dear Reviewer H4Yh, thanks for your time reviewing our paper. We have meticulously prepared a detailed response addressing the concerns you raised. Could you please have a look to see if there are further questions? Your invaluable input is greatly appreciated. Thank you once again, and we hope you have a wonderful day!

---

> ### Author Response · Authors · 2023-11-21
> **Could you please have a look at our rebuttal?**
>
> Dear Reviewer H4Yh, thanks for your time reviewing our paper. We have meticulously prepared a detailed response and updated manuscript addressing the concerns you raised. Could you please have a look to see if our response has resolved your concerns? Your invaluable input is greatly appreciated. Once again, thank you, and we wish you a wonderful day!

---

> > ### Comment · Reviewer_H4Yh · 2023-11-21
> >
> > Dear authors,
> >
> > Thank you for your careful responses and revision to the paper. I will be sure to read them soon and consider my rating.
> >
> > Best,\
> > Reviewer H4Yh

---

> > > ### Author Response · Authors · 2023-11-23
> > > **Your feedback is invaluable to us**
> > >
> > > Dear Reviewer H4Yh, thanks for your reply! Could you kindly reconsider the rating? Your feedback is invaluable to us. Thank you once again, and we hope you have a wonderful day!

---

### Official Review · Reviewer_MKb4 · 2023-11-09

**Soundness:** 3 good
**Presentation:** 3 good
**Contribution:** 3 good
**Rating:** 6
**Confidence:** 4

**Summary:**

- The paper investigates the causes and mechanisms behind robust overfitting in adversarial training (AT). Robust overfitting refers to the phenomenon where a model's robust test accuracy declines as training progresses.
- Through factor ablation experiments, the authors show that the factors inducing robust overfitting originate from the normal training data, not the adversarial perturbations.
- They explain robust overfitting as arising from the model learning "non-effective" features that lack robust generalization. Specifically, due to distributional differences between training and test data, features that are robust on training data may not generalize to be robust on test data.
- As training progresses, the gap between the model's learning states on training vs test data widens. This facilitates the proliferation of non-effective features. When optimization is dominated by these features, robust overfitting occurs.
- Based on this understanding, the authors propose two measures to regulate the learning of non-effective features: 1) Attack strength - using higher attack budgets to eliminate non-effective features. 2) Data augmentation - to align the model's learning state on training and test data.
- Experiments show clear correlations between the extent of robust overfitting and the degree to which these measures suppress non-effective features. The proposed methods mitigate robust overfitting and improve adversarial robustness across different models and datasets.
- Overall, the work provides an explanation of robust overfitting from the perspective of features and learning states. The understanding and analysis seem quite intuitive and comprehensive. The paper makes a valuable contribution towards demystifying the mechanisms behind this phenomenon.

**Strengths:**

Originality: The paper provides a novel perspective on understanding robust overfitting by treating normal data and perturbations as separate factors. The idea of non-effective features that lack robust generalization is also an original concept proposed in this work.

Quality: The study is scientifically rigorous, with principled factor ablation experiments that isolate the effect of normal data. The analysis and explanations are intuitive yet comprehensive. The proposed methods demonstrate consistent effectiveness.

Clarity: The paper is very clearly written and structured. The background provides sufficient context. The experiments and results are well-described. The analysis logically builds up an explanation of robust overfitting in an easy to follow manner.

Significance: Robust overfitting is a major impediment in adversarial training that lacks a satisfactory explanation. This work makes significant headway by unraveling its underlying mechanisms. The insights can inform the design of more effective defenses. Overall, this is an impactful study that advances fundamental understanding of an important phenomenon in adversarial machine learning.

In summary, the originality of the conceptual framing, rigorous experimental methods, clear writing, and significance of the research problem make this a compelling paper with multiple strengths. It is a valuable contribution that sheds light on the mechanisms behind robust overfitting through a meticulous and insightful analysis.

**Weaknesses:**

- While effective, the attack strength and data augmentation measures may not represent optimal or sufficient solutions. More advanced techniques informed by this analysis could further enhance robustness.
- The theoretical analysis relies on intuitive reasoning. Formalizing the notions of robust/non-robust features and quantifying the learning state gap could strengthen the conceptual framing.
- The focus is on explaining robust overfitting, less on maximizing robust accuracy. Follow-up work could build on these insights to achieve state-of-the-art robustness.
- The experiments primarily use simple CNN architectures on CIFAR datasets. Testing the analysis on larger datasets and SOTA models could reveal additional insights.
- There is limited ablation on the proposed methods themselves. Varying their hyperparameters and components could better isolate their effects.
- The writing could further improve clarity in some areas, like explicitly defining "small-loss" data earlier on.

**Questions:**

1. The analysis relies on the notion of a "gap" between training and test learning states. Is there a principled way to quantify this gap? Are there any theoretical bounds on the gap size that induces robust overfitting?
2. Have you experimented with more advanced data augmentation techniques like MixUp or CutMix? Could these further help with state alignment and reducing non-effective features?
3. How well do your insights transfer to larger scale problems like ImageNet? Are there any key differences in robust overfitting that you observe in such settings?

---

> ### Author Response · Authors · 2023-11-17
> **Response to Reviewer MKb4**
>
> We thank the reviewer for the positive and constructive comments regarding our paper.
>
> **Q1:** Notion of learning state gap.\
> **A1:** The term "learning state gap" between the training and test sets has been revised to "robustness gap" in our updated manuscript. This change has been made consistently throughout the paper. Additionally, we employ quantitative metrics such as adversarial loss and robust accuracy to measure the model's robustness, providing evidential support for the existence of a robustness gap between the training and test datasets, as illustrated in Figure 1(b). Regarding the process of how the robustness gap induces robust overfitting, our analysis is as follows: during the adversarial training process, the model's adversarial robustness on the training set exceeds that on the test set, as evidenced by the adversarial loss and robust accuracy observed in Figure 1(b). This results in a robustness gap between the training and test data. Considering that adversarial perturbations are generated on-the-fly and adaptively adjusted based on the model's robustness. The robustness gap between the training and test data leads to distinct adversarial perturbation on the robust features in these datasets. The varying degree of adversarial perturbation on the robust features in the training and test data amplify the distribution differences between these two datasets, thereby degrading the generalization of robust features on the training set and facilitating the generation of non-effective features. The increasing non-effective features further exacerbates the robustness gap between training and test data, forming a vicious cycle. These analyses are detailed in our updated manuscript presented in Section 1, Section 3, and Figure 2. We appreciate the valuable suggestion to refine our analysis, and we believe this adjustment enhances the clarity and rigour of our research.
>
> **Q2:** Different data augmentation techniques.\
> **A2:** Indeed, we conducted experiments with four different data augmentation techniques, as shown in Figure 1(d) and Figure 6, all of which demonstrated similar effects on robust overfitting. This consistency arises from the fact that data augmentation techniques are applied with a targeted transformation objective to modify the proportion of small-loss adversarial data, as outlined in Algorithm 2. Consequently, they inherently achieve the effect of robustness alignment, and the specific data augmentation techniques employed do not significantly impact their effectiveness.
>
> **Q3:** Large-scale dataset like ImageNet.\
> **A3:** Due to constraints on computational resources, training adversarially robust models on a large-scale dataset like ImageNet is currently beyond our capacity. As far as I am aware, the phenomenon of robust overfitting is predominantly investigated on standard benchmark datasets such as CIFAR10. I regret to admit my uncertainty regarding potential differences in the robust overfitting phenomenon between the CIFAR10 and ImageNet datasets.
>
> Thank you once again for your valuable comments, which have helped us improve the clarity and rigour of our research. Hope our elaborations and the revisions made in the new manuscript could address your concerns. Please let us know if there is more to clarify.

---

### Author Response · Authors · 2023-11-17
**Summary of Revision**

Dear Reviewers,

We sincerely appreciate the time and effort you have dedicated to reviewing our manuscript. We have carefully considered your valuable suggestions and made substantial revisions to our work. The main changes are summarized below:

1. Added a flowchart to illustrate the onset of robust overfitting, as shown in Figure 2.
2. Replaced all instances of "learning state gap" with "robustness gap".
3. Provided evidence supporting the robustness gap between the training and test datasets in Figure 1 (b).
4. Updated the analysis of how the robustness gap between the training and test sets prompts the generation of non-effective features in Sections 1 and 3.
5. Provided detailed experimental hyperparameters in Table 3, as suggested by Reviewer BeRL.
6. Included the pseudocodes of the proposed methods in Algorithm 1 and Algorithm 2, as suggested by Reviewer BeRL and Reviewer H4Yh.
7. Moved the analysis of existing techniques for mitigating robust overfitting to the appendix to alleviate the reading burden, as suggested by Reviewer 6tzX.
8. Highlighted the best results in Table 2, as suggested by Reviewer 6tzX.
9. Corrected the error in Eqn (4), as suggested by Reviewer H4Yh.

Once again, we express our gratitude for your valuable feedback, which have helped us improve the clarity and rigour of our research. Hope our elaborations and the revisions made in the new manuscript could address your concerns. Please let us know if there is more to clarify.

Sincerely,\
The authors

---

### Author Response · Authors · 2023-11-22
**Summary of Logical Framework**

In order to more easily grasp the contributions of our work, we summarize the logical framework of our work as follows:

**Research problem:** The onset of robust overfitting, which is an important problem in adversarial training and its fundamental mechanism is not fully understood.

**Motivation:** We conducted a series of rigorously factor ablation experiments following the principles of the controlled variable method, inferring that the factors inducing robust overfitting originate from normal data. Normal data can be considered as a composition of features, which motivate us to investigate the onset of robust overfitting from the perspective of feature generalization.

**Intuitive analysis:** We propose the concept of non-effective features, which are the robust features in the training set that lack generalization. Further, we analyze in detail how non-effective features are generated during the adversarial training process: The model has a robustness gap between the training and test datasets. Considering that adversarial perturbations are generated on-the-fly and adaptively adjusted based on the model’s robustness. The robustness gap between the training and test datasets leads to distinct adversarial perturbation on the robust features in these datasets, thereby degrading the generalization of robust features on the training set and facilitating the generation of non-effective features. The increasing non-effective features further exacerbates the robustness gap between training and test datasets, forming a vicious cycle. This, in turn, gives rise to the phenomenon of robust overfitting.

**Verification methods:** We propose two orthogonal methods to validate our analysis. Specifically, attack strength regulates the model's learning of non-effective features by eliminating non-effective features on the training dataset. Data augmentation controls the generation of non-effective features by aligning the robustness of the model on the training and test datasets. Extensive experiments verify our analysis of robust overfitting.

**Robustness evaluation:** The proposed approaches consistently enhance the adversarial robustness of adversarial training methods across a wide range of experimental settings, demonstrating their effectiveness. Besides, they can also contribute to the complementary robustness improvement on the comparison with advanced robust overfitting relief technologies, highlighting the significance of our analysis to robust overfitting.

**Highlights:** We propose a new perspective to explain the onset of robust overfitting. To the best of our knowledge, our analysis is the first to explain this empirical behaviors of robust overfitting: after the learning rate decays in adversarial training, why the model's robustness in test dataset first rises for a short period of time and then continues to decline.

---

### Meta-Review · Area_Chair_8Tbm · 2023-12-04

**Metareview:**

I have read all the materials of this paper including the manuscript, appendix, comments, and response. Based on collected information from all reviewers and my personal judgment, I can make the recommendation on this paper, *reject*. No objection from reviewers who participated in the internal discussion was raised against the reject recommendation.

**Summary**

Reviewer MKb4 did a good job on the paper summarization (Other reviewers' comments are also very good; MKb4 is the first reviewer in the display). I omitted mine here.

**Research Question**

The authors aim to discover the fundamental mechanism behind robust overfitting.

**Motivation & Challenge Analysis**

I did not a specific challenge of the above research question. As far as I can see, there is no work investigating the onset of robust overfitting.

**Philosophy or Style**

Since there is no specific challenge, there is no philosophy to tackle the challenge, either. The style of this paper is mainly based on empirical intuition.

**Concerns**

1. Presentation is a major issue of this paper. (a) When I read the abstract, the information is overloaded, or a burden to me. I can understand every sentence, but I failed to summarize the key idea of this paper in one or two sentences. I am not sure whether the current version can be separated into several papers. (b) Some sentences are very long over 3 lines. (c) Some notations are used without formal definitions, such as "effective features." Effective to what? The word "effective" is too general. (d) I suggested a figure or a diagram to illustrate the key idea in the introduction part. (e) Difficulty in understanding figures. For example, what is the experimental setting in Figure 1. Figure 1 (c-d) come before their introduction, which cause the unnecessary confusion. The caption of Figure 2 is too short that I cannot read it. What order? What do the arrows meaning? What do the light blue lines mean? (f) Some typos, such as "we analysis" in the abstract should be "we analyze"; "a detail analysis" in the fourth paragraph of introduction should be "a detailed analysis."

2. Logic. Some logics are not solid to me. (a) In Section 3.1, the authors followed the data ablation principle and designed three experiments, *baseline*, *data & perturbation*, and *perturbation*. Why focusing on the small-loss adversarial data? What is the performance of *data only*? The authors try to convince readers to follow their path, but I am thinking about whether there exist other factors, for example, the interaction between *data*  and *perturbation*. (b) The authors would like to trace the factor from the sample-level to the feature level. This is a good idea, but why doing this? Maybe in a separate paper? I have to talk about the above presentation point again. The authors have their own path without informing the readers, so that every part is strange. Admitted, every part is related but not coherent. (c) The authors proposed two measures to verify their findings. Why the procedure is correct and suitable is not illustrated. Why two measures? Not one or three? (d) Some findings are derived from empirical intuition. How to generalize to other datasets? I understand large datasets like ImageNet take huge computational resources, but there are still several mid-size datasets, such as Animal, Food, Dog and so on, for extensive evaluations.

3. Evaluation. Since this paper follows the empirical intuition style, some guidance for practical usage is necessary. For example, parameter analysis, different losses or architectures, diverse tasks.

**Justification For Why Not Higher Score:**

This paper is not self-standing and does not reach the bar of ICLR.

**Justification For Why Not Lower Score:**

N/A

---

### Decision · Program_Chairs · 2024-01-16

Reject